# BENCHMARKING MLLMS ON TOPOLOGICAL REASONING OF CHEMICAL REACTION DIAGRAMS

## ABSTRACT

Chemical reaction diagrams are visual representations of complex process graphs, where understanding the overall pathway, including its branches, cycles, and flow, is crucial. While Multimodal Large Language Models (MLLMs) have shown proficiency in recognizing the individual nodes of these graphs, such as molecules and reagents, their ability to perform topological reasoning on the entire structure remains critically underexplored. This creates an urgent need for a targeted evaluation framework to probe this higher-order skill. Fulfilling this need, this paper introduces a systematic benchmark to evaluate this specific capability. We present **ReactBench**, a collection of 1,618 question-answer pairs designed to measure MLLM performance on a hierarchy of tasks, from component recognition to complex topological analysis. Our evaluation of state-of-the-art models reveals a significant deficit: while GPT-4o achieves 79.71% accuracy on node-level identification tasks, its performance plummets to 49.5% on questions that require true topological reasoning about the pathway. By providing the first focused benchmark for this skill, our work establishes a rigorous methodology for diagnosing a key failure mode in MLLMs and guiding the development of models that can comprehend the full, structured processes depicted in scientific diagrams.

## 1 INTRODUCTION

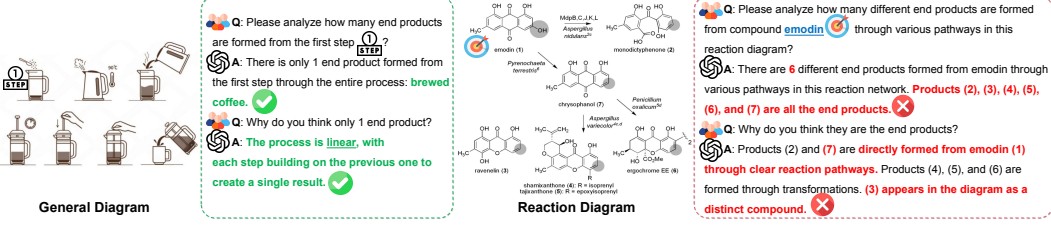

Figure 1: **Comparing General Diagram and Chemical Reaction Diagram on Element Identification Tasks.** GPT-4o performs well on general diagrams for identification tasks. However, for chemical reaction diagram, the structural complexity of molecular diagrams and reaction pathways leads to errors.

Recent advances in multimodal large language models (MLLMs) have demonstrated remarkable capabilities in visual-linguistic reasoning across diverse domains (OpenAI, 2024b; Anthropic, 2024; Team et al., 2024; Bai et al., 2025). However, their capacity for structured visual reasoning, particularly topological analysis of complex diagrams, remains underexplored. This limitation is especially pronounced in chemical domains, where a wide array of diagrams involving molecular representations, reaction pathways, and experimental visualizations requires sophisticated spatial reasoning capabilities (Qian et al., 2023; Masry et al., 2022). Among these challenging visual formats, chemical reaction diagrams represent particularly information dense and topologically complex structures that demand advanced graph theoretic reasoning (Garcia et al., 2024; Shah et al., 2024).

The core challenge in topological reasoning extends far beyond simple pattern recognition, requiring models to understand spatial relationships, connectivity patterns, and hierarchical abstraction from local visual features to global structural properties. This challenge is illustrated through a

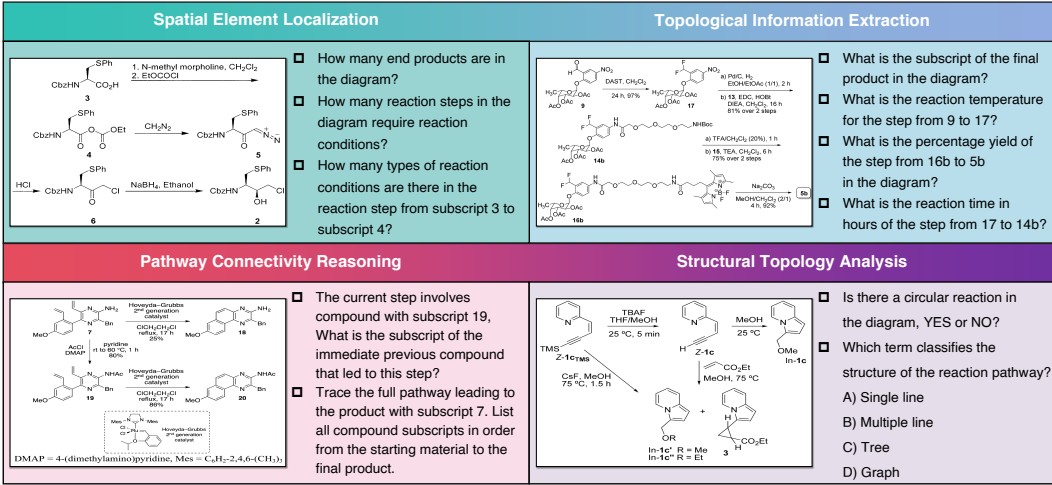

Figure 2: **ReactBench evaluation framework.** ReactBench systematically evaluates chemical reasoning across four core dimensions. Tasks are stratified by complexity to comprehensively evaluate the topological reasoning capabilities of chemical reaction diagrams.

controlled comparison of topological reasoning complexity in Fig. 1. When presented with simple linear topologies (sequential processes like coffee brewing), GPT-4o demonstrates robust path tracing capabilities and correctly identifies the single terminal node. However, when analyzing complex branching topologies with multiple pathways in chemical reaction networks, the same model exhibits systematic failures in distinguishing intermediate vertices from sink nodes, incorrectly classifying six endpoints instead of the actual three terminal products. This reveals a fundamental limitation in graph theoretic reasoning capabilities, where models excel at local visual pattern recognition but fail at global structural understanding. This limitation raises fundamental questions about current MLLMs' structural reasoning capabilities: ***How effectively do MLLMs perform hierarchical abstraction from local visual features to global topological properties in complex diagrams?*** For instance, while models may correctly identify individual molecular structures (local features), they often fail to classify the overall reaction network as a linear chain versus a branching tree structure (global topology). ***Can current architectures maintain structural consistency during multi-hop reasoning over branching graph topologies?*** For example, when tracing a pathway from reactant A through intermediate B to product C, models may lose track of alternative branches from B, incorrectly concluding that C is the only possible outcome from A.

To systematically evaluate these capabilities, we argue that models must be assessed on visual spatial representations directly. Existing approaches that convert diagrams to text formats like SMILES through Optical Chemical Structure Recognition (OCSR) are inadequate for topological reasoning evaluation. Such text-based representations discard crucial spatial positioning, pathway directionality, and global connectivity patterns that are essential for structural analysis. Moreover, OCSR transcription errors would further degrade reasoning performance, making it impossible to isolate and evaluate pure topological reasoning capabilities.

While the need for direct structural reasoning is clear, existing benchmarks that evaluate MLLMs on document understanding and visual question answering primarily focus on semantic comprehension rather than structural reasoning capabilities. To address this gap, we introduce **ReactBench**, a systematic benchmark for evaluating topological reasoning in MLLMs through chemical reaction diagrams. ReactBench isolates topological reasoning as a distinct and measurable capability, featuring hierarchical task decomposition from local spatial localization to global structural classification. We have carefully curated 1,618 question-answer pairs from real-world scientific literature and patents, spanning four complementary dimensions that systematically probe different aspects of visual spatial reasoning.

Our main contributions are:

- We introduce ReactBench, a systematic benchmark for evaluating topological reasoning in MLLMs, featuring hierarchical task decomposition from spatial component localization to structural graph classification, comprising 1,618 expertly annotated question answer pairs across four skill dimensions.

- We establish empirical evidence for a fundamental capability gap in current MLLMs: while they excel at local visual pattern recognition, they exhibit systematic failures in graph theoretic reasoning and structural abstraction, with performance gaps of up to 40% between local recognition and global structural analysis tasks.

- Through controlled evaluation across multiple model families, we identify specific failure modes in multi-hop reasoning and hierarchical abstraction, providing quantitative baselines and concrete insights for improving visual spatial reasoning capabilities in future multimodal architectures.

ReactBench provides a systematic evaluation framework for assessing topological reasoning capabilities in MLLMs. Our findings suggest that current models face challenges in integrating local visual processing with global structural understanding, highlighting areas for future improvement in multimodal architectures.

## 2 RELATED WORK

### 2.1 MULTIMODAL LARGE LANGUAGE MODELS

MLLMs extend text-only LLMs by incorporating visual comprehension capabilities. Early works like CLIP (Radford et al., 2021) established vision-language alignment through contrastive learning. Recent MLLMs mainly follow two design paradigms. The first includes encoder-decoder architectures like BLIP-2 (Li et al., 2023) that use separate visual and textual encoders with a Q-Former (Zhang et al., 2023) for cross-modal fusion. The second encompasses decoder-only architectures, such as LLaVA Liu et al. (2023), that project visual features directly into the LLM input space. These models are typically trained through a multistage process involving pre-training on image-text pairs and instruction tuning on high-quality multimodal datasets. Recent advances include KOSMOS-1 Huang et al. (2023), which supports multimodal in-context learning, and Qwen2.5-VL Bai et al. (2025) and GPT-4V OpenAI (2024a), which demonstrate strong performance in various vision language tasks, including visual reasoning and OCR understanding.

### 2.2 CHEMICAL REACTION DIAGRAM UNDERSTANDING

To position our ReactBench in the context of existing chemical reaction diagram understanding efforts, we first review prior work and then highlight how our benchmark departs from theirs. While RxnScribe Qian et al. (2023), ReactionDataExtractor 2.0 Wilary & Cole (2023) provide valuable element detection capabilities, they do not assess holistic semantic understanding of reaction sequences. While prior work like RxnScribe and ReactionDataExtractor 2.0 has focused on component detection and element extraction from reaction diagrams, ReactBench addresses a fundamentally different challenge: evaluating MLLMs' capacity for topological reasoning and structural understanding through comprehensive question-answering tasks that require integrating visual perception with graph-theoretic analysis.

## 3 THE REACTBENCH BENCHMARK

### 3.1 OVERVIEW

We introduce ReactBench, a specialized benchmark designed to evaluate the ability of MLLMs to comprehend and reason about chemical reaction diagrams. ReactBench consists of a high-quality dataset ChemReaction with 1,618 rigorously curated QA pairs derived from real-world chemical reaction diagrams, and a comprehensive evaluation framework assessing four key dimensions of reaction diagram understanding: (1) spatial element localization (2) topological information extraction (3) pathway connectivity reasoning (4) structural topology analysis. Key statistics of ChemReaction are summarized in Tab. 1, and its composition is illustrated in Fig. 3. The complete benchmark construction pipeline is depicted in Fig. 4. Below, we detail the dataset collection and annotation methodology for ChemReaction, followed by the evaluation scope of ReactBench.

| Category | Number |
|---|---|
| **Question Dimensions** | |
| Spatial Element Localization | 835 (52%) |
| Topological Information Extraction | 414 (26%) |
| Pathway Connectivity Reasoning | 167 (10%) |
| Structural Topology Analysis | 202 (12%) |
| **Question types** | |
| Numerical questions | 1094 (67%) |
| Multiple-choice questions | 202 (13%) |
| Free-form questions | 322 (20%) |
| **Total** | **1618** |

Table 1: Key statistics of ChemReaction.

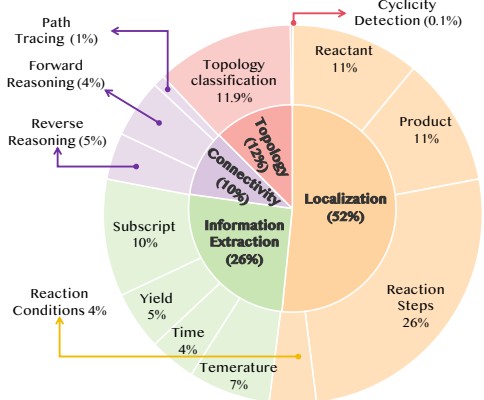

Figure 3: Composition of ChemReaction.

## 3.2 DATASET COLLECTION

ChemReaction is constructed from a collection of 1,300+ reaction diagrams systematically curated from chemical literature and patent databases, ensuring both academic rigor and industrial relevance. Through a multi-stage filtering process that excludes diagrams with incomplete mechanisms, ambiguous visual representations, or duplicate content, we obtain a high-fidelity dataset integrating diverse layout structures with nuanced mechanistic details. To capture the full spectrum of visual complexity and reasoning challenges in reaction diagrams, the curated subset is stratified into four categories representing real-world workflows: (**a**) *Single-line*: Linear pathways with unidirectional flows; (**b**) *Multiple-lines*: Parallel or branched mechanisms with competing intermediates; (**c**) *Tree*: Hierarchical branching with divergent synthesis routes; (**d**) *Graph*: Cyclic networks requiring non-linear reasoning. This taxonomy covers prevalent reaction topologies across experimental and industrial contexts with balanced coverage.

## 3.3 QUESTION DESIGN

To enable a detailed evaluation of MLLMs' capabilities in understanding chemical reaction diagrams, as shown in Fig. 2, we have systematically designed questions across four key dimensions:

**Spatial Element Localization.** This task evaluates models' capacity for fine-grained spatial reasoning and object localization within structured diagrams. Given a reaction network represented as a visual graph, models must accurately identify and classify node types (source, sink, intermediate) and edge properties (directed connections, branching points) based solely on topological position and visual context. This requires robust visual grounding and spatial attention mechanisms without reliance on domain-specific semantic priors.

**Topological Information Extraction.** This dimension assesses models' ability to perform structured information extraction guided by topological context. Models must extract numerical and textual attributes associated with specific graph elements while maintaining spatial correspondence between visual tokens and their semantic roles. The task challenges models' capacity for fine-grained visual-linguistic alignment and context-dependent parsing in dense, multi-modal representations.

**Pathway Connectivity Reasoning.** This dimension probes models' grasp of connectivity and path traversal in reaction networks. Models must trace pathways, predict sequential nodes, and reconstruct complete paths based purely on topological connections. These tasks demand graph-theoretic reasoning, minimizing the need for predictive chemical knowledge.

**Structural Topology Analysis.** This dimension focuses on models' ability to classify and analyze the overall topological structure of reaction diagrams. Models must identify structural patterns (linear, branched, cyclic, tree-like, or graph topologies) and detect structural properties like cycles or terminal nodes. This requires holistic graph-theoretic understanding of diagram architecture.

Figure 4: **ReactBench dataset construction and evaluation pipeline.** (a) Data acquisition involves systematic collection of chemical reaction diagrams from peer-reviewed literature and patent databases, followed by preprocessing through manual curation and quality filtering protocols. (b) Annotation framework encompasses structured generation and expert validation of question-answer pairs, ensuring alignment with target evaluation dimensions and chemical accuracy. (c) Systematic evaluation protocol for assessing multimodal large language model performance across predefined reasoning dimensions with comprehensive analysis metrics.

## 3.4 ANNOTATION AND VALIDATION

All questions and answers are meticulously annotated by multiple chemistry experts to ensure accuracy and consistency. The annotation process involves several iterative rounds of cross-checking, during which discrepancies are discussed and resolved collaboratively. This rigorous multi-round review mechanism guarantees high-quality annotations, providing a robust benchmark for evaluating MLLMs' understanding of chemical reaction diagrams.

## 3.5 EVALUATION METHOD

To simplify the evaluation process, all questions in ChemReaction are presented in either multiple-choice or open-ended formats with concise, easily verifiable ground truth answers. We utilize prompting and template matching to extract answers. Prompts guide the model in generating responses in both full and short-answer formats. After generations, the short answer is extracted to compare it with the ground truth. Detailed prompts used in our experiments can be found in the Appendix A. This design eliminates the need for using MLLMs as judges (Chen et al., 2024a) and enables fully automated accuracy measurement through exact string matching.

## 4 EXPERIMENT

### 4.1 EXPERIMENTAL SETUPS

In our experiments, we comprehensively evaluate various MLLMs across multiple chemical reaction diagram understanding tasks. We test open-source models including Qwen2.5-VL Bai et al. (2025) (3B, 7B, 72B), InternVL2.5 Chen et al. (2024b) (8B, 8B-MPO, 26B, 78B), LLaVA-NeXT Liu et al. (2024) (Mistral-7B, Vicuna-13B), MiniCPM-o 2.6 Yao et al. (2024) (8B), DeepSeek-VL2 Wu et al. (2024) (Tiny and Small) and Phi-3.5-vision-instruct Abdin et al. (2024) (4B). Furthermore, we evaluate API-based models such as GPT-4o OpenAI (2024b), Claude-3.5-Sonnet Anthropic (2024), Gemini-1.5-Pro Team et al. (2024), and Qwen-VL-Max Qwen (2024). Evaluation of these MLLMs ensures fair comparison considering model architecture differences and parameter scales.

### 4.2 MAIN RESULTS

In this section, we extensively evaluate both open-source and API-based MLLMs on our React-Bench. Tab. 2 summarizes the performance of various models across four key task dimensions: Spatial Element Localization, Topological Information Extraction, Pathway Connectivity Reasoning, and Structural Topology Analysis. Despite the rapid advancements in MLLMs, our results re-

| Models | # Params | ReactBench | | | | Average |
|---|---|---|---|---|---|---|
| | | Localization | Extraction | Reasoning | Analysis | |
| **API-based Models** | | | | | | |
| GPT-4o | - | 44.07 | 79.71 | 82.04 | 49.50 | 63.83 |
| Claude-3.5-Sonnet | - | **50.78** | 81.88 | **92.22** | 49.50 | **68.60** |
| Gemini-1.5-Pro | - | 45.75 | 75.60 | 71.86 | **55.45** | 62.17 |
| Qwen-VL-MAX | - | 49.22 | **90.10** | 88.62 | 32.18 | 65.03 |
| **Open Source Models** | | | | | | |
| InternVL2.5-MPO | 8B | 38.20 | 66.43 | 59.28 | 37.62 | 50.38 |
| InternVL2.5 | 8B | 27.43 | 65.22 | 45.51 | 26.73 | 41.22 |
| InternVL2.5 | 26B | 33.89 | 66.18 | 65.87 | 19.80 | 46.44 |
| InternVL2.5 | 78B | 41.08 | 79.47 | 67.66 | 26.73 | 53.74 |
| Qwen2.5-VL | 3B | 24.43 | 89.61 | 49.70 | 34.65 | 49.60 |
| Qwen2.5-VL | 7B | 37.96 | 85.75 | 66.47 | 39.11 | 57.32 |
| Qwen2.5-VL | 72B | **46.47** | **89.86** | **83.83** | **54.46** | **68.66** |
| DeepSeek-VL2 | 3B | 45.39 | 50.72 | 16.17 | 33.66 | 36.49 |
| DeepSeek-VL2 | 16B | 28.62 | 78.50 | 8.38 | 28.71 | 36.05 |
| LLaVA-NeXT-Mistral | 7B | 18.44 | 30.43 | 26.95 | 21.78 | 24.40 |
| LLaVA-NeXT-Vicuna | 13B | 13.77 | 39.13 | 1.80 | 28.22 | 20.73 |
| Phi-3.5-vision-instruct | 4.2B | 21.80 | 69.75 | 29.34 | 38.12 | 39.75 |
| MiniCPM-o 2.6 | 8B | 35.09 | 75.12 | 51.50 | 39.60 | 50.33 |

Table 2: Detailed evaluation results on ReactBench across different models, showing the answer accuracy (Acc.) of each model in each task. **Average** denotes the arithmetic mean of a model's scores over all four tasks. The **best** and second-best per model category is highlighted with bold and underlined respectively.

veal that these models lack the ability to comprehend complex reaction sequences and overarching structural patterns in chemical diagrams.

**Results of Spatial Element Localization.** This task assesses the models' ability to identify key elements in reaction diagrams. Despite its simplicity, most models perform worst here, highlighting challenges with chemical notation. API-based models lead, with Claude-3.5-Sonnet (Acc. 50.78%) and Qwen-VL-MAX (Acc. 49.22%) ranking highest. Larger models ($\geq$13B) generally outperform smaller ones (<7B), suggesting that model capacity influences performance. However, inconsistencies emerge: InternVL2.5-26B (Acc. 33.89%) trails its 8B version (Acc. 38.20%), and LLaVA-NeXT-Vicuna-13B obtains the lowest accuracy, showing that size alone isn't decisive. Benchmark confirms MLLMs struggle with chemical diagrams, underscoring need for domain-specific training.

**Results of Topological Information Extraction.** This task evaluates the models' ability to extract detailed textual and numerical information. While most models perform well on this task (except for GPT-4o and Gemini-1.5-Pro), Qwen-VL-MAX leads with Acc. 90.1%, followed closely by Qwen2.5VL-72B (Acc. 89.86%). Notably, some API-based models with over 13B parameters score lower than smaller models like Qwen2.5VL-3B (Acc. 89.61%) and Qwen2.5VL-7B (Acc. 85.75%), indicating that specific architectural optimizations can significantly enhance performance. In stark contrast, LLaVA-NeXT-Mistral-7B only obtains 30.43% at Acc., highlighting the limitations of lower-parameter models in handling the complex details of chemical reaction diagrams. Results highlight need for specialized strategies to enhance MLLMs in this domain.

**Results of Pathway Connectivity Reasoning.** This task tests the models' ability to correctly order compounds and reaction steps in a chemical mechanism, requiring multi-step inference and causal reasoning. Closed-source models like Claude-3.5-Sonnet (Acc. 92.22%) and Qwen-VL-MAX (Acc. 88.62%) lead the performance, while some large models like Deepseek-VL2-Small (Acc. 16.17%) and LLaVA-NeXT-Vicuna-13B (Acc. 1.8%) perform very poorly. Although larger models sometimes show improved ordering with increased parameters, differences in training strategies also play a significant role. Overall, many models that excel at information extraction struggle with sequence reasoning, highlighting the need for more specialized training to better understand reaction order.

**Results of Structural Topology Analysis.** This task evaluates the models' ability to interpret the overall topology of reaction diagrams. The overall performance is weak. Among API-based models, Gemini-1.5-Pro (Acc. 55.45%) scores highest, while Qwen-VL-MAX (Acc. 32.18%) falls behind despite strong extraction abilities. Open-source models vary widely; for instance, Qwen2.5VL-72B (Acc. 54.46%) leads, but others like InternVL2.5-26B (Acc. 19.8%) score poorly. Smaller models also struggle, though some mid-sized ones outperform larger peers, indicating that architecture and

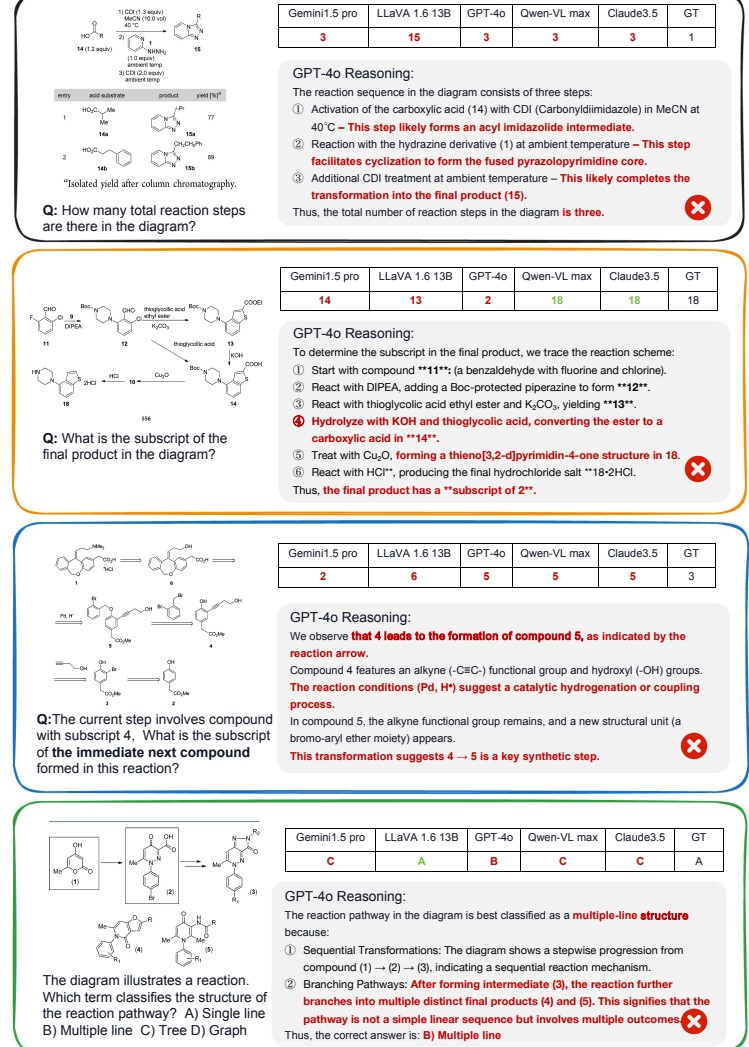

Figure 5: Qualitative analysis of recurring failure patterns in ReactBench evaluation. Each case study exemplifies a characteristic reasoning deficit observed across multiple model architectures.

training data are more crucial than size. These findings underscore the need for domain-specific improvements in global structural reasoning.

**Overall Performance and Discussion.** No model excels consistently across all tasks. Among open-source options, Qwen-VL-MAX leads overall. While some models handle element identification or information extraction well, they falter in sequence reasoning and structure analysis—revealing the multifaceted challenges of interpreting reaction diagrams. Robust visual parsing, precise extraction, and strong reasoning are essential. Significant gaps in extraction and structure analysis necessitate domain-specific knowledge and enhanced image-text alignment, motivating specialized domain-adaptive models.

## 5 DIAGNOSING THE REASONING GAP: FROM QUALITATIVE FAILURES TO CORE LIMITATIONS

While our benchmark results quantify a significant performance deficit, a deeper analysis is required to understand the root causes of these failures and provide actionable insights for the field. This section, therefore, moves from demonstrating that models fail to investigating why they fail. We present a diagnostic analysis that first identifies the consistent symptoms of flawed topological reasoning through qualitative case studies. Subsequently, we conduct a controlled experiment to isolate the root cause, demonstrating that the critical bottleneck is a fundamental limitation in the models' core reasoning capabilities, not merely a failure of visual perception.

## 5.1 Symptom Analysis: A Pattern of Failure in Global Reasoning

The following qualitative examples are not random errors; they are consistent symptoms of a core deficiency. They reveal a recurring pattern where models excel at local pattern recognition but fail to integrate these perceptions into a coherent global understanding.

**Failure in Hierarchical Abstraction (Confusing Local Conditions with Global Steps):** As shown in Row 1 of Fig. 5, MLLMs correctly identify local textual markers (e.g., 1), 2), 3)) but cannot perform the hierarchical abstraction needed to understand these are conditions for a single global transformation. This leads to a systematic over-segmentation of the process, revealing an inability to move from local feature recognition to global process understanding.

**Loss of Structural Consistency in Multi-Hop Reasoning:** Tracking entities requires reasoning across multiple steps. In Row 2, GPT-4o correctly follows the main molecular transformation but is derailed by a local, irrelevant visual feature ($2HCl$). This demonstrates a breakdown in structural consistency, as the model fails to maintain focus across the hops of a reaction pathway.

**Failure in Topological Path Tracing:** MLLMs often disregard explicit topological rules in favor of learned heuristics. In Row 3, models ignore the directed arrow ($5 \rightarrow 4 \rightarrow 3$) and instead predict the next step based on a superficial numerical sequence. This highlights a failure of graph theoretic reasoning—an inability to follow the diagram's fundamental connective logic.

**Inability to Classify Global Graph Topology:** The ultimate test of structural reasoning is classifying a diagram's global topology. In Row 4, models correctly perceive local features like a linear sequence and a branching point, but fail the hierarchical abstraction required to classify the global "single line" structure. This directly confirms our thesis: local perception is intact, but global structural classification fails.

## 5.2 Isolating the Root Cause: Perception vs. Reasoning

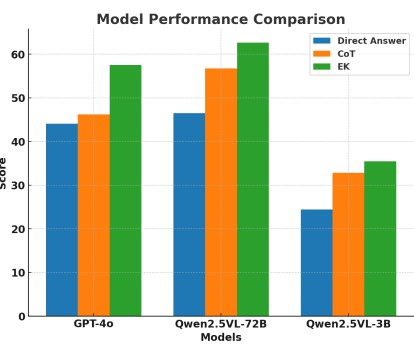

Figure 6: Performance comparison of three models with two enhancement techniques: Chain-of-Thought (CoT) and External Knowledge (EK).

The preceding symptoms consistently show MLLMs failing to reason about global structure. But does this stem from an inability to see the diagram correctly (a perception error) or an inability to think about it correctly (a reasoning deficit)? To definitively answer this question, we design experiments to isolate the reasoning component by controlling the quality of the structural information provided to the models.

First, we explore whether guiding the models to structure their own perception could enhance their reasoning. Drawing on Chain-of-Thought (CoT) methodologies (Wang et al., 2025; Wei et al., 2022; Sprague et al., 2024; Deng et al., 2024), we prompt models to first decompose the chemical diagram into a structured JSON representation. This structured data is then used to answer the question. As illustrated in Fig. 6, this structured perception step modestly improves identification performance for GPT-4o, Qwen2.5VL-72B, and Qwen2.5VL-3B, with gains of 2.16%, 8.38%, and 10.30%, respectively. While this shows that enforcing a more systematic visual analysis is beneficial, the gains are limited, suggesting the problem runs deeper than disorganized perception.

To eliminate the perception variable entirely, we then supply the models with perfect, error-free structural information. We manually annotate each image with its ground-truth structured data, providing this as External Knowledge (EK) to assist the model's reasoning (see Appendix A.5). This experimental setup bypasses the model's own visual processing and tests its reasoning capabilities in a best-case scenario.

The results, shown in Fig. 6, are striking. This approach leads to significant performance gains, with accuracies reaching 57.49% for GPT-4o and 62.63% for Qwen2.5VL. However, this is our most critical finding: even when a model is given flawless, structured knowledge of every component and connection in the diagram, its accuracy remains below 65%.

This proves that the primary bottleneck is not the visual perception front-end but a deep-seated limitation in the MLLM's core reasoning capabilities. The fact that performance is still severely constrained, even with perfect input, underscores that simply improving visual recognition will be insufficient to solve these complex, domain-specific tasks.

## 5.3 THE ROLE OF CHEMICAL VISUAL CONTEXT IN TOPOLOGICAL REASONING

We further investigate whether the reasoning bottleneck stems from topological analysis itself or from the integration between visual processing and structural reasoning. By replacing molecular structures with black borders, as illustrated in Appendix A.1, we examine how models perform when deprived of familiar visual prompts.

As shown in Tab. 3, masking molecular structures leads to consistent performance drops across all task categories, including those requiring chemical knowledge (Localization, Extraction, Reasoning) as well as the Analysis task that focuses on topological structure. While most models show modest declines in Analysis performance (-3.46% for 3B, +1.48% for 72B), one model (7B) exhibits a notable improvement (+8.41%).

| Model | Task | Original images | Masked images | $\Delta$ |
|---|---|---|---|---|
| Qwen2.5-VL-3B | Localization | 24.43 | 19.76 | -4.67 |
| | Extraction | 89.61 | 78.50 | -11.11 |
| | Reasoning | 49.70 | 42.51 | -7.19 |
| | Analysis | 34.65 | 31.19 | -3.46 |
| Qwen2.5-VL-7B | Localization | 37.96 | 35.81 | -2.15 |
| | Extraction | 85.75 | 77.78 | -7.97 |
| | Reasoning | 66.47 | 56.89 | -9.58 |
| | Analysis | 39.11 | 47.52 | +8.41 |
| Qwen2.5-VL-72B | Localization | 46.47 | 45.15 | -1.32 |
| | Extraction | 89.86 | 80.68 | -9.18 |
| | Reasoning | 83.83 | 68.26 | -15.57 |
| | Analysis | 54.46 | 55.94 | +1.48 |

Table 3: Impact of molecular placeholder masking on model performance.

These results suggest that even topological analysis benefits from chemical visual context in most cases. The predominant pattern of performance decline indicates that models rely on molecular structures not merely for chemical identification, but as visual anchors that help organize their understanding of reaction connectivity and flow. The inconsistent results across model sizes suggest that the relationship between visual complexity and reasoning performance may depend on model capacity and training specifics.

Rather than indicating a clear separation between chemical and topological reasoning, these findings point to the integrated nature of visual processing and structural analysis in chemical diagram understanding. Future training approaches should account for this interdependence, focusing on how models can effectively leverage chemical visual elements to support various types of reasoning tasks.

## 5.4 KEY FINDINGS AND IMPLICATIONS FOR FUTURE WORK

Our diagnostic analysis provides clear evidence that addresses the questions from our introduction and offers concrete directions for the field, directly highlighting our technical contribution.

**The Bottleneck is Reasoning, Not Just Perception:** Our central finding is that the challenge of structured visual analysis is fundamentally a reasoning problem. Simply improving visual encoders or OCR will yield diminishing returns. Even with perfect perception, current architectures lack the sophisticated reasoning skills to interpret complex topologies.

**A Core Deficit in Hierarchical Abstraction:** We provide empirical evidence that MLLMs systematically fail at hierarchical abstraction. They get lost in local details and cannot build a coherent global understanding from constituent parts. This is a crucial, underexplored failure mode that our benchmark is specifically designed to measure.

## 6 CONCLUSION

We introduce ReactBench, a benchmark to systematically investigate MLLMs' comprehension capabilities in chemical reaction diagrams. We evaluate MLLMs using 1,618 QA pairs based on real-world reaction images, covering tasks from basic recognition to advanced reasoning. Our results reveal significant limitations in MLLMs' ability to integrate visual and chemical knowledge, particularly in complex, multi-step reactions. Our work highlights the need for improved multimodal reasoning in scientific domains and provides a critical evaluation framework for the community.

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

## A    DETAILED EXPERIMENT SETUPS

In this section, we provide more details about our experiment designs.

### A.1    ILLUSTRATION OF THE ABLATION STUDY

To provide a clear visual example of the ablation study described in the main paper, Fig. 7 illustrates how chemical diagrams are modified.

### A.2    PROMPTS FOR RESPONSE GENERATION

In our experiments, we prompt the MLLMs to generate responses to different types of questions, such as multiple choice, number, and text types. The prompts used for these question types are shown in Tab. 4.

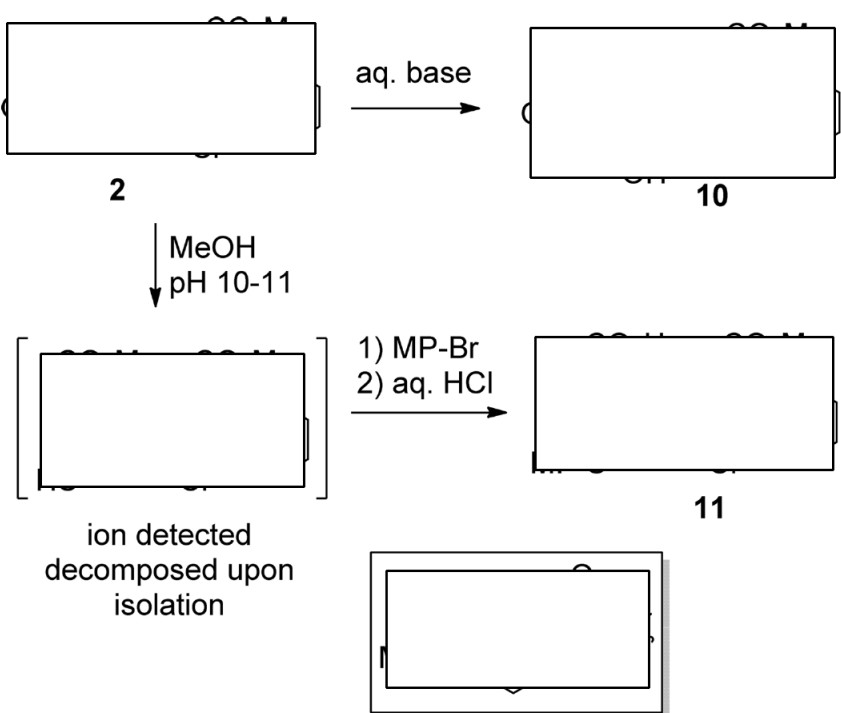

Figure 7: **Illustration of the ablation study**: replacing molecule images with rectangle placeholders

| Answer type | prompt |
|---|---|
| multiple choice | Just provide the corresponing choice option, such as 'A', 'B', 'C', or 'D'. |
| number | Provide ONLY the numerical answer without any units, symbols, or additional text. |
| text | Just give the subscript without any other text. |

Table 4: The prompt for different QA types in answer generation.

### A.3 PROMPT FOR ANSWER EXTRACTION

In our main experiment, we directly pose questions to the model and employ prompt engineering and template matching to extract answers. As illustrated in Fig. 8, prompts guide the model in generating responses in both full and short answer formats, where the short answer format adheres to the requirements specified in Tab. 4 for different question types. After generation, the short answer is extracted for comparison with the ground truth, while the full answer is used to analyze the reasoning process of MLLMs.

### A.4 PROMPT OF CHAIN-OF-THOUGHT

The prompt of the CoT prompting strategy is illustrated in Fig. 9, where the model is first instructed to convert the reaction diagram into a predefined JSON format, followed by the question being posed.

### A.5 PROMPT OF EXTERNAL KNOWLEDGE

Here, we present the specific prompt and external knowledge format used in the main text. An illustrative example is provided in Fig. 10 for reference.

## B    EXAMPLES IN REACTBENCH

In this section, we present selected samples from different tasks in our dataset, as illustrated in Fig. 12, 13, 14, and 15. These examples highlight the diversity of our designed questions, which encompass a wide range of problem types.

---

**Direct Answer System Prompt**

**Multimodal Chemical Reaction QA Task**
**Inputs**:
1. Chemical reaction scheme image
2. Question

**Answer the question by following these rules:**
1. Provide **ONLY** the numerical answer (e.g., \"80\") without any units, symbols, or additional text.
2. Provide a detailed explanation of the step-by-step reasoning.\n"

**Final Output Format (as a JSON object):**
```
{
  "   \"answer\": \"<numerical_value>\",
  "   \"explanation\": \"<step-by-step reasoning>\"
}
```

Figure 8: **The direct answer prompt in our ReactBench.**

---

**Chain of Thought System Prompt**

**Chemical Reaction QA Task:**
**"Step 1 - Structural Parsing: Analyze the reaction image and convert it to this JSON format:"**
```
[{
   \"reactants\": [
        "{"      \"category\": \"[Mol]\","
              \"category_id\": 1,"
              \"bbox\": [x_min, y_min, x_max, y_max],"
              \"compound_id\": \"...\"}]
        //Additional reactants..."],

     \"conditions\": [
        "{"    \"category\": \"[Txt]\","
              \"category_id\": 2,"
              \"bbox\": [x_min, y_min, x_max, y_max],"
              \"text\": [\"reagent\", \"(equiv)\"]"}"
         / Additional conditions..."],

     \"products\": [
         // Product entries...]
     // "Additional reactions..."
     ]
```
**"Step 2 - Question Answering:"**
Using the parsed structure from Step 1, answer the question by following these rules:"
1. Provide ONLY the numerical answer (e.g., \"80\") without any units, symbols, or additional text."
2. Provide a detailed explanation of the step-by-step reasoning."

**"Final Output Format (as a JSON object): "**
```
   "{"
       \"answer\": \"<numerical_value>\","
       \"explanation\": \"<step-by-step reasoning>\""
   "}"
```

Figure 9: **The prompt of Chain-of-Thought in our ReactBench.**

---

## C    THE USE OF LARGE LANGUAGE MODELS

An LLM (ChatGPT) was used only for grammar checking and stylistic polishing; it did not contribute to research ideas, methods, experiments, code, figures, or analyses. All outputs were reviewed and edited by the authors.

External Knowledge System Prompt

**Multimodal Chemical Reaction QA Tas**k
   **Inputs**:
      1. Chemical reaction scheme image
      2. Supplemental JSON data

   **Task Requirements:**
      1. Cross-validate information between image and JSON data
      2. Answer format requirements:
         - Return ONLY numerical value (e.g. \"80\")
         - No units, symbols or additional text"
      3. Explicitly explain how both modalities contribute to the answer

   **Output Format (strict JSON):**
   {
     \"answer\": \"<numerical_value>\","
      \"explanation\": \"<integration_steps>\""
   }

Figure 10: **The prompt of external knowledge in our ReactBench.**

| entry | catalysts | $T$ (°C) | $t$ (h) | conv.$^g$ (%) | select.$^g$ (%) | | |
|-------|-----------|----------|---------|---------------|------|------|------|
|       |           |          |         |               | 2 | 3 | 4 |

{
  "bboxes": [
        { "id": 0, "bbox": [ 829.38, 21.2,   204.02, 163.45], "category_id": 1}, // molecules
        { "id": 1, "bbox": [ 1079.75, 26.38, 204.02, 132.37 ], "category_id": 1}, // molecules
        { "id": 2, "bbox": [ 466.77, 5.66, 284.32, 178.12 ], "category_id": 1 }, // molecules
  { "id": 3, "bbox": [ 7.14, 275.7, 1333.64, 150.9 ], "category_id": 4 },
        { "id": 4, "bbox": [1210.98,203.37,27.9,34.8],"category_id": 3 },
        { "id": 5, "bbox": [ 60.14, 12.57, 285.17, 146.18], "category_id": 1}, //molecules
        { "id": 6, "bbox": [ 229.35, 203.37, 22.72,35.67 ], "category_id": 3},
        { "id": 7, "bbox": [ 640.31, 200.78, 27.9, 35.67], "category_id": 3},
        { "id": 8, "bbox": [ 964.06, 210.27, 25.31, 34.81], "category_id": 3
        }],
  "reactions": [ { "reactants": [5], "conditions": [], "products": [2,0,1]} ]
}

Figure 11: **External Knowledge Example in our ReactBench.** This JSON represents molecular structures and reaction relationships extracted from the image. The `bboxes` section defines detected elements with their respective IDs, bounding box coordinates, and the `category_id`, where 1 indicates molecules, 2 denotes text, 3 corresponds to compound identifiers (numerical labels without molecular structures), and 4 represents auxiliary information. The `reactions` section associates reactants and products by referencing their IDs, illustrating the transformations occurring between molecules. Overall, it systematically organizes visual and chemical data for structured interpretation.

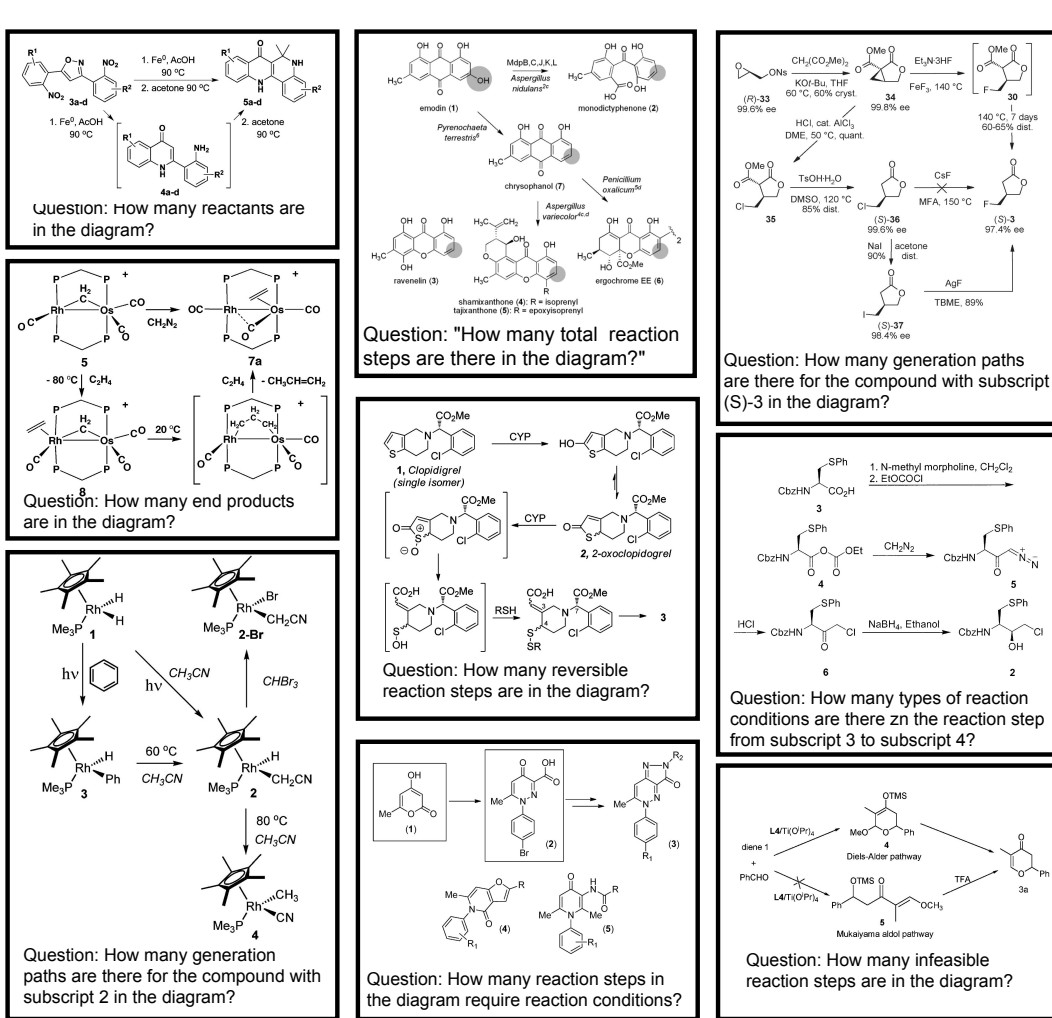

Figure 12: **Examples of Element Identification Task in our ReactBench.**

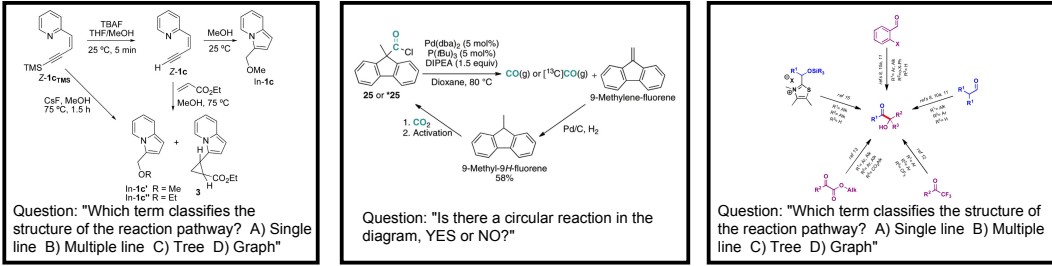

Figure 13: **Examples of Structure Analysis Task in our ReactBench.**

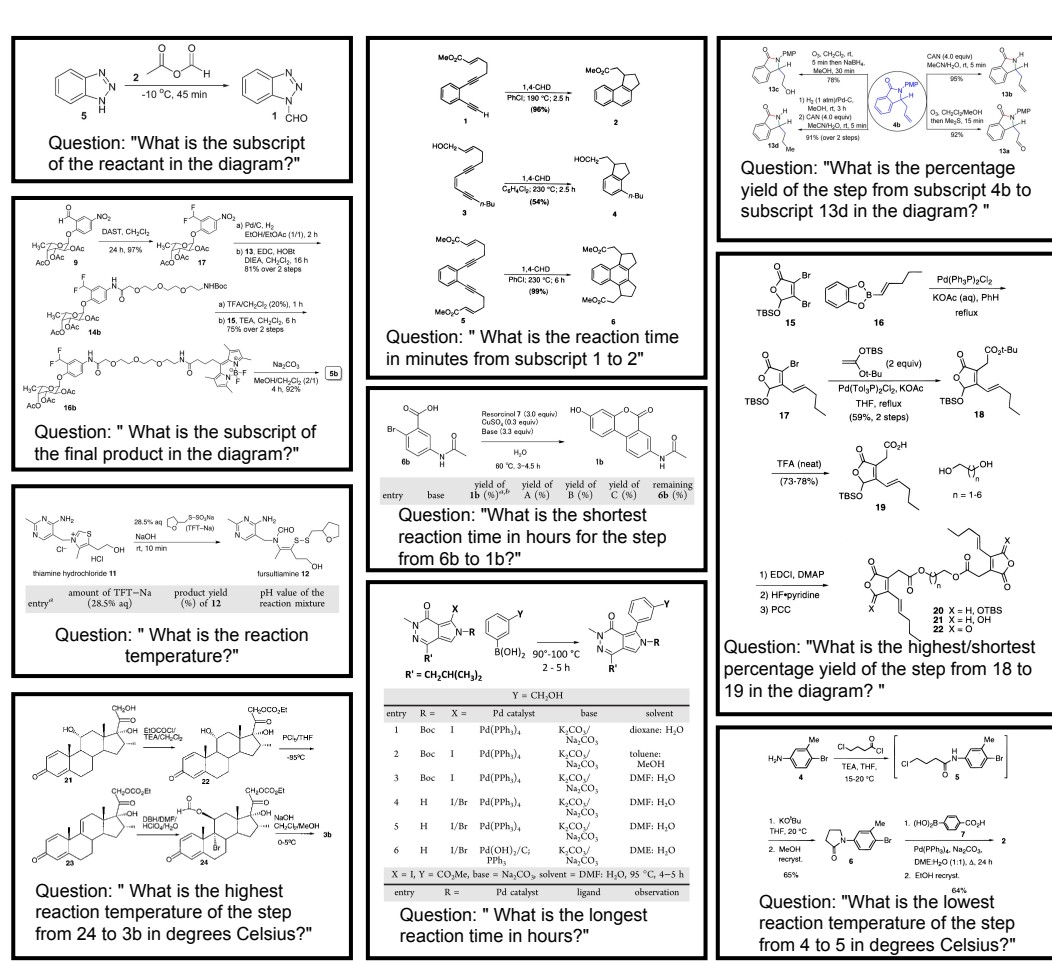

Figure 14: **Examples of Information Extraction Task in our ReactBench.**

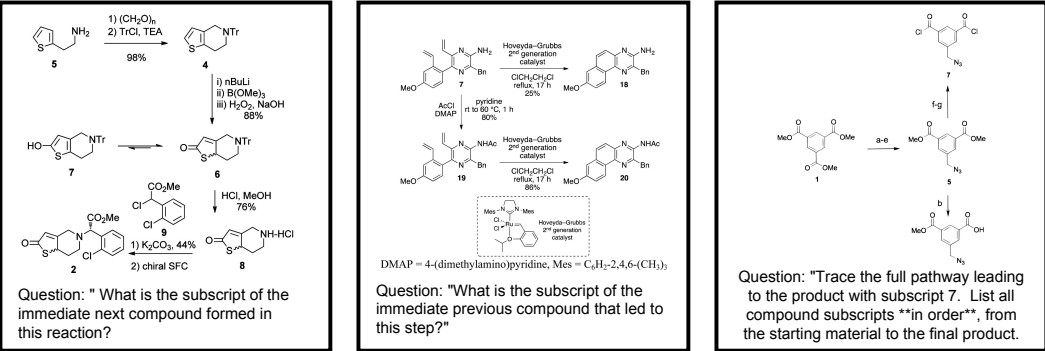

Figure 15: **Examples of Sequence Reasoning Task in our ReactBench.**

