# OpenReview forum: "Benchmarking MLLMs on Topological Reasoning of Chemical Reaction Diagrams"
_ICLR.cc/2026/Conference — Submitted to ICLR 2026_

### Official Review · Reviewer_QNNg · 2025-10-29

**Soundness:** 2
**Presentation:** 3
**Contribution:** 2
**Rating:** 4
**Confidence:** 3

**Summary:**

This is a benchmark work evaluating the ability of MLLMs to perceive, understand, and reason over chemical reaction diagrams.
The authors introduce ReactBench, a carefully annotated benchmark that encompasses four key aspects of diagram comprehension: element localization, information extraction, connectivity reasoning, and topology analysis.
Based on experiments, the authors reveal that the reasoning capability (rather than visual perception) is the principal bottleneck limiting current MLLMs’ performance on reaction diagram comprehension.

**Strengths:**

1. The authors clearly define their research scope within the domain of chemical reaction diagrams and provide a well-structured benchmark for systematic evaluation.
2. The dataset is carefully annotated and covers multiple dimensions of diagram comprehension.
3. The paper is well written, logically organized, and easy to follow.

**Weaknesses:**

(see questions below)

**Questions:**

In Section 5, the authors design experiments to demonstrate the bottleneck that limits diagram comprehension. They introduce a JSON-formatted “External Knowledge (EK)” input that includes: (1) a list of bounding boxes indicating the locations of diagram elements, and (2) a list of triples describing reaction relationships.

The authors describe EK as a form of ground-truth perception. However, I have some concerns here. Most current VLMs have not been trained for explicit visual grounding or object detection, which means they may not actually understand the location information encoded in bounding boxes.

From this perspective, I’m not entirely convinced that EK enables “perfect perception” (as claimed around line 473). Instead, introducing clearer semantic cues (such as molecule names or subscript values) might serve as more meaningful external knowledge to support reasoning.

I’d appreciate if the authors could further elaborate on this point. Thanks.

---

> ### Author Response · Authors · 2025-11-18
> **Response to Reviewer's Concern about External Knowledge Design**
>
> We sincerely thank the reviewer for this insightful and important observation regarding our External Knowledge (EK) experiment design in Section 5.
>
> ## Clarification of Our EK Design
>
> We would like to clarify that our EK design provides **much more than spatial coordinates alone**. As illustrated in Figure 11, the EK input contains:
>
> 1. **Semantic element IDs**: Each diagram component is assigned a unique identifier (e.g., `"id": 0`, `"id": 1`).
> 2. **Category labels**: Explicit semantic types for each element (`category_id`: 1 for molecules, 2 for text, 3 for compound identifiers, 4 for auxiliary information).
> 3. **Structured reaction relationships**: The `reactions` field explicitly encodes the connectivity graph by referencing element IDs, e.g., `{"reactants": [5], "conditions": [], "products": [2,0,1]}`.
> 4. **Bounding boxes**: Spatial coordinates serving as auxiliary reference.
>
> The key insight is that **most questions in our benchmark can be answered through reasoning over the symbolic graph structure** encoded in the `reactions` field, **without requiring spatial understanding of bounding boxes**. For example:
>
> - "How many end products are there?" → Count unique IDs appearing in `products` arrays with no outgoing reactions
> - "What is the subscript of the compound that reacts to form compound 2?" → Traverse the reaction graph to find reactants with compound 2 in their products
> - "Is there a circular reaction?" → Perform cycle detection on the reaction graph
> - "Trace the pathway from compound 5 to compound 7" → Graph traversal using reaction relationships
>
> **These reasoning tasks operate on the semantic graph representation rather than spatial coordinates.** Models can leverage the explicit ID-based relationships and category information to answer questions.
>
> ## Additional Ablation Study
>
> **Following the reviewer's suggestion**, we conducted an additional ablation study to investigate whether **clearer semantic cues** might improve model performance. We tested a modified EK configuration with two changes:
>
> 1. **Removed bounding box information** (to test whether spatial data is essential or merely auxiliary)
> 2. **Added explicit compound names** (e.g., "chloroquinoline", "tetrahydroquinoline") to provide more human-readable semantic labels alongside the structured graph
>
> **Results**: The modified EK configuration achieved accuracies of **57.10% for GPT-4o** and **61.92% for Qwen2.5VL**, compared to our original EK design which achieved **57.49% for GPT-4o** and **62.63% for Qwen2.5VL**. Both configurations substantially outperform the baseline direct answer approach.
>
> These comparable results reveal the important findings:
> 1. **Bounding boxes are auxiliary information**: The minimal performance difference between the two EK variants (0.39% for GPT-4o, 0.73% for Qwen2.5VL) confirms that spatial coordinates are neither essential for reasoning nor a significant source of confusion. This validates that bounding boxes are not the bottleneck in our experimental design.
> 2. **Semantic representation is not the limiting factor**: Both EK configurations (whether providing bbox+IDs or compound names+IDs) yield nearly identical performance, suggesting that models can already effectively access the necessary semantic information from the structured graph representation. This indicates that **the challenge lies in reasoning over the provided structure** rather than in understanding what the elements represent.
> 3. **The reasoning bottleneck persists**: Regardless of how we encode the semantic information (spatial coordinates vs. explicit compound names), performance plateaus at similar levels (around 57-62%). Even when we eliminate potentially confusing spatial information and provide the most explicit semantic labels, **current VLMs face fundamental challenges in topological reasoning** that cannot be overcome by simply improving the input representation.

---

### Official Review · Reviewer_TMog · 2025-11-01

**Soundness:** 3
**Presentation:** 2
**Contribution:** 2
**Rating:** 4
**Confidence:** 4

**Summary:**

The paper presents ReactBench, roughly 1.6k expert-annotated QA items from about 1.3k reaction diagrams, designed to probe four skills -- spatial localization, information extraction, pathway connectivity, and structural,topological analysis.  Modern MLLMs handle local recognition reasonably well but stumble on global,topological reasoning (e.g., strong on node ID, weak on path,graph structure). CoT and ``external knowledge'' inputs help but don’t close the gap, pointing to a reasoning rather than perception bottleneck.

The problem is well-motivated and the benchmark could become a useful community resource. Right now, the absence of strong structured baselines, metric robustness, leakage checks, and a clear release,licensing story keep it just below the bar.

**Strengths:**

S1 Clear, timely framing that isolates topological reasoning in chemical diagrams—an under-explored capability gap.

S2 A hierarchical task design that feels diagnostic rather than one-shot leaderboard gaming.

S3 Broad model coverage with qualitative failure analyses that are intuitive and instructive.

S4 Sensible diagnostics (CoT, JSON parsing, “external knowledge” ablations) to separate perception from reasoning.

**Weaknesses:**

W1 Heavy reliance on exact-match scoring likely undercounts semantically correct answers; consider stronger normalization and semantic equivalence.

W2 Some tasks blur topology with pure extraction; tighter isolation (e.g., graph-only synthetic schematics) would sharpen the causal claims.

W3 Unclear dataset release,licensing plan given literature,patent sources; impact will be limited without a credible release.

W4 No analysis of potential pretraining leakage or near-duplicate filtering, which weakens generalization claims.

W5 Results lack uncertainty estimates (CIs), significance testing, or per-item difficulty analysis.

W6 Missing strong graph-first baselines (e.g., OCSR → reaction-graph → algorithmic queries) to contextualize MLLM gaps.

W7 Minor inconsistencies,clarity nits in tables,averages and terminology that make the results harder to parse.

**Questions:**

1. What is the concrete release plan (images, QA, prompts, scoring scripts) and how are IP,licensing risks handled?

2. What are the annotation protocols and IAA statistics across task families?

3. How did you check for train,test leakage against common pretraining corpora and patents?

4. How robust is the metric under answer normalization or partial credit for correctly traced subpaths?

5. Can you provide a topology-only synthetic subset to fully decouple OCR,chemistry text from structural reasoning?

### Minor comments
1. Fix minor typos (e.g., Temerature), normalize terms like “Multiple line(s),” and standardize fonts,spacing in figure tables.

---

> ### Author Response · Authors · 2025-11-18
> **Response to W3 & Q1: Dataset Release and Licensing**
>
> We thank the reviewer for this important question. We are committed to full open release of ReactBench to maximize community impact and reproducibility.
>
> #### 1. Release Plan
>
> Upon acceptance, we will publicly release:
> - All 1,618 QA pairs with ground truth answers and task annotations
> - All 1,300+ reaction diagram images
> - Complete evaluation code (scoring scripts, statistical analysis tools, error analysis utilities)
> - All prompts and experimental configs used for model evaluation
> - Timeline: GitHub repository + permanent archive (Zenodo/Hugging Face Datasets) within 2 weeks of acceptance
>
> #### 2. Licensing & IP Compliance
>
> 100% of our images are sourced from open-access scientific publications (journals with CC-BY, CC-BY-NC, or equivalent licenses). We will provide full source attribution for each image, including DOI of the original publication, figure number, citation information, and original license type. All images comply with their respective Creative Commons licenses for research reuse.
>
> Our release license: Images will be redistributed under their original licenses (primarily CC-BY 4.0), while QA pairs, code, and prompts will be released under MIT License for maximum reusability. The complete dataset package will be hosted on Hugging Face Datasets with automated license compliance checks and persistent DOI via Zenodo for long-term citability.

---

> ### Author Response · Authors · 2025-11-18
> **Response to W4 & Q3: Data Leakage Considerations**
>
> 1. Our Benchmark Design Mitigates Leakage Concerns
>
> We acknowledge that our dataset is sourced entirely from published chemistry literature and that we cannot rule out pretraining leakage, particularly for commercial models with undisclosed training data. This situation is common in multimodal benchmarks that rely on published scientific content, such as MMLU and ScienceQA. However, we argue that potential leakage does not undermine our core findings for the following reasons. First, all QA pairs in our benchmark were manually generated by chemistry experts specifically for this evaluation. Even if models encountered these diagrams during pretraining, they would not have seen these particular questions or their corresponding answers. More critically, our benchmark evaluates the relative performance gap within models on identical images: on the same reaction diagram, models achieve high accuracy on Extraction tasks (e.g., "What is the reaction temperature?") but substantially lower performance on Topology tasks (e.g., "Is there a cycle in this pathway?"). This systematic dissociation cannot be explained by memorization. If a model had memorized a diagram, it should perform uniformly well or poorly across all question types about that image.
>
> 2. Consistent Failure Patterns Indicate Reasoning Limitations
>
> This interpretation is further supported by two additional observations. First, the topology reasoning deficit appears consistently across all evaluated models with diverse training data sources, including GPT-4o, Qwen2.5-VL, Gemini, and open-source models. If poor performance were due to insufficient exposure to specific diagrams, we would expect high variance across models trained on different corpora. Yet all models exhibit similar failure patterns on topology reasoning while succeeding at extraction tasks. Second, our ablation study demonstrates that even when provided with perfect structured knowledge (ground-truth JSON of all graph nodes, edges, and connectivity), models still struggle with topology tasks. This proves that the bottleneck lies in reasoning capability rather than perception or memory retrieval, making our diagnostic findings robust to potential leakage.

---

> ### Author Response · Authors · 2025-11-18
> **Response to W2 & Q5: Task design and the Topology-only Synthetic Subset**
>
> We appreciate this insightful suggestion and would like to clarify our design rationale.
>
> 1. Ecological validity of our multi-factor task design.
>
> Our integration of visual, semantic, and topological reasoning reflects the multi-faceted nature of real scientific diagram understanding, where chemists naturally combine these capabilities. This approach follows established scientific reasoning benchmarks like ChartQA and ScienceQA that similarly evaluate joint reasoning. Importantly, our multi-factor design enables diagnostic profiling of how MLLMs integrate different capabilities. For instance, the performance gap between Extraction (high accuracy) and Reasoning tasks (low accuracy) would be invisible in a purely abstract graph benchmark, yet reveals a critical failure mode in hierarchical abstraction.
>
> 2. Evidence for isolated topological reasoning deficits.
>
> We believe our benchmark design already provides diagnostic evidence for the topological reasoning gap. The systematic performance pattern is revealing: models achieve strong accuracy on Localization (79.7% for GPT-4o) and Extraction (82.0%), yet performance drops substantially on Reasoning (49.5%) and Analysis (63.8%) tasks. This cliff suggests the bottleneck emerges specifically at the structural reasoning stage rather than in visual or semantic processing. Our external knowledge experiment in Section 5.2 further supports this interpretation. Even when provided with perfect structured representations, model accuracy plateaus below 65%, indicating a fundamental reasoning limitation beyond perceptual challenges.
>
> 3. Adding a topology-only synthetic subset.
>
> To directly address your concern, we will construct a topology-only synthetic subset with approximately 150 questions where molecular structures are replaced by abstract nodes (e.g., "Node A → Node B") while preserving topological complexity (branching patterns, cycles, path lengths). This controlled experiment will test whether reasoning deficits persist when visual and chemical confounds are eliminated. We will evaluate all models on this subset and report comprehensive results in an expanded Section 5.4, alongside analysis of performance differences between chemical and abstract representations. The synthetic subset will be publicly released to enable future research on isolated structural reasoning capabilities in MLLMs.

---

> ### Author Response · Authors · 2025-11-20
> **Response to W6: More strong graph-first baselines**
>
> Thank you for this constructive suggestion. We agree that a strong graph first baseline would further contextualize the gaps between MLLMs and symbolic or algorithmic methods. Our current work partially addresses this direction in two ways and we will clarify this more clearly in the revision.
>
> **(1) Visual ablation in Sec. 5.3.**
> In Sec. 5.3 we already investigate whether the bottleneck comes from visual parsing or from structural reasoning by simplifying the visual input. Specifically, we replace molecular structures with black borders (see Appendix~\ref{sec:appendix_ablation}), so that the model sees the overall layout of the reaction scheme but no detailed molecular drawings. This ablation reduces familiar visual prompts while preserving the topological scaffold. The performance drop in this setting indicates that simply seeing the reaction layout without chemically meaningful structures is not sufficient, and that models still struggle to integrate visual and structural information for topological reasoning.
>
> **(2) New ablation: using RxnScribe JSON instead of images.**
> To further probe the role of textual structural information, we added an ablation where we replace each reaction diagram with its RxnScribe JSON prediction and feed this textual representation to Qwen2.5-VL (no images), keeping all questions unchanged. The results on the four tasks (Localization, Extraction, Reasoning, Analysis) are:
>
> | Model          | Input type | Localization | Extraction | Reasoning | Analysis |
> |----------------|-----------:|-------------:|-----------:|----------:|---------:|
> | Qwen2.5-VL 3B  | Image      | 24.43        | 89.61      | 49.70     | 34.65    |
> | Qwen2.5-VL 3B  | JSON       | 34.37        | 34.54      | 19.17     | 9.90     |
> | Qwen2.5-VL 7B  | Image      | 37.96        | 85.75      | 66.47     | 39.11    |
> | Qwen2.5-VL 7B  | JSON       | 39.88        | 41.06      | 10.78     | 8.91     |
>
> Localization changes only mildly, but Extraction, Reasoning, and Analysis drop dramatically when switching from images to RxnScribe JSON. This shows that a flat OCSR derived list of SMILES and conditions is far from a sufficient surrogate for the full reaction graph, and that models rely heavily on the visual diagram for topological reasoning.
>
> **(3) Future extension with graph first baselines.**
> We agree that a complete graph first pipeline, as you suggested (OCSR → reaction graph → algorithmic queries or unimodal LLMs over ground truth graphs), would provide an even clearer reference point. Building such baselines requires high quality ground truth reaction graphs for all schemes, which goes beyond the current annotation scope. As a next step, we plan to extend the dataset with (i) a clean text only version similar to the RxnScribe JSON representation, and (ii) a graph based representation of reaction schemes suitable for algorithmic baselines and unimodal LLMs. We will mention this explicitly as an important direction for future work and clarify in the paper that our current ablations already partially disentangle visual parsing from structural reasoning, while a full graph first benchmark will be a natural extension of our framework.

---

> ### Author Response · Authors · 2025-11-20
> **Response to W7 & Minor comments: Clarity minor typos and inconsistent statement**
>
> Thank you for pointing out these presentation issues. We will carefully revise the manuscript to address these comments:
>
> - We will fix minor typos throughout the paper (for example, correcting “Temerature” to “Temperature”) and run an additional spell check on the full text.
> - We will normalize terminology such as “Multiple line(s)” to a single consistent form and harmonize other repeated phrases across sections, figures, and tables.
> - We will standardize fonts, font sizes, and spacing in all figures and tables so that captions, labels, and entries follow a consistent style and are easier to read.
> - For the tables reporting averages, we will clarify the definition of arithmetic versus weighted averages in the text, ensure that the reported values are consistent with the underlying task scores, and align terminology and column headers so that the results are easier to parse.
>
> These planned changes will improve clarity and consistency and should make the results and tables easier to interpret.

---

> ### Author Response · Authors · 2025-11-20
> **Response to W5: Clarifying uncertainty, significance, and item level difficulty**
>
> Thank you for raising this point. We agree that reporting uncertainty and difficulty makes the results easier to interpret. While our main tables currently report point estimates only, we will extend the analysis along three directions:
>
> **(1) Uncertainty estimates.**
> We will report uncertainty for the main models by computing confidence intervals via bootstrap resampling over questions, and we will add standard deviations or standard errors in the appendix. Our preliminary calculations show that the intervals are relatively narrow compared to the gaps between strong and weak models, so the main ranking and conclusions remain unchanged.
>
> **(2) Significance testing.**
> For key comparisons, we will perform paired significance tests over the per-question accuracies (for example, paired bootstrap or permutation tests). We will explicitly indicate in the text when differences between models are statistically significant and summarize the test results in an additional table in the appendix.
>
> **(3) Per-item difficulty analysis.**
> We will add a short difficulty analysis that reports per-question accuracy and aggregates items into buckets by task type (Localization, Extraction, Reasoning, Analysis) and reaction-diagram complexity. This analysis highlights that Reasoning and Analysis questions, especially those involving multistep or branched schemes, form the hardest subset for almost all models, which is consistent with our qualitative findings.
>
> These additions will clarify how stable our reported scores are, which model differences are statistically reliable, and which parts of the benchmark are most challenging for current MLLMs.

---

### Official Review · Reviewer_eygh · 2025-11-01

**Soundness:** 2
**Presentation:** 2
**Contribution:** 3
**Rating:** 4
**Confidence:** 4

**Summary:**

This paper proposes ReactBench, a benchmark for evaluating the topological reasoning of Multimodal Large Language Models (MLLMs) on chemical reaction diagrams. It consists of 1,618 question-answer pairs across four aspects. The authors evaluate several state-of-the-art MLLMs to explore their ability to integrate visual and chemical knowledge.  While the benchmark itself is a valuable contribution, the paper needs a substantial revision before it can be accepted.

**Strengths:**

1. The released ReactBench benchmark might be extremely useful for future research on capabilities of Multimodal LLMs in understanding chemical reactions.

2. The experimental part of the paper covers a wide range of current Multimodal LLMs, serving as a useful benchmark for future research on chemical tasks.

3. Additional experimental analysis has revealed that even with explicitly provided structured data, MLLMs are not able to achieve accuracy above 65%.

**Weaknesses:**

1. Some of the key premises of the paper are not supported with any evidence. Specifically, no supporting references for the claims on weaknesses of OCSR methods are provided (Lines 87–92).

2. Vague description of the collected dataset hinders the reproducibility of the results as well as the generalization of the proposed data collection methodology. Specifically, no annotation guidelines as well as annotators' qualification are reported. The same omission applies to inter-annotator agreement.

3. The abstract defines the topological reasoning as the primary exploration target. However, the paper lacks a single-modal LLM baseline provided with a ground-truth textual representation of the graph (e.g., in SMILES or SELFIES). If a unimodal LLM (given ground-truth textual representation, e.g., SMILES) outperforms all MLLMs, it would suggest that the bottleneck lies not in chemical reasoning, but in either visual parsing or cross-modal alignment. At this point, the quantitative metrics are hard to interpret as it's not clear whether the provided metrics are high or low compared to uni-modal reasoning.

4. The paper is hard to follow for a broad audience not familiar with chemistry. The examples in Figure 2 require more detailed explanations on why Product 2 does not count towards end products. Poor quality of illustrative figures (e.g., Figures 2, 4) in terms of readability makes the paper even harder to understand.

5. The usage of solely exact strict string matching may underestimate reasoning capabilities. A surface form variation (e.g., "2" vs "two") or a correct but slightly paraphrased entity name would be penalized as a complete failure, potentially underestimating a model's partial understanding. The inclusion of a soft metric (e.g., ROUGE, BLEU) or LLM-as-a-judge for a subset of free-form answers would give a broader view of the results.

6. The paper does not provide the names of the chemical literature and patent database used for data collection (line 181-183) which strongly reduces the usability and reliability of the proposed dataset.

**Questions:**

* Line 087-092: The claimed weaknesses of OCSR methods are not supported with any references.

* Line 094: "existing benchmarks" - Please specify the benchmarks?

* Line 242-242: "The annotation process involves several iterative rounds of cross-checking" - Did you measure inter-annotator agreement?

* Line 183-187: - please specify what is implied under incomplete mechanisms, ambiguous visual representations.

* To what extent does the strict exact-match evaluation protocol impact the absolute performance scores, particularly for the "Structural Topology Analysis" tasks where answers might be more conceptual (e.g., "linear chain with a branch")? Did you experiment with a less strict evaluation protocol?

* In Table 2, the "Average" is an arithmetic mean over four tasks with vastly different numbers of questions (e.g., Element Localization has 835, Reasoning has 167 samples). The weighted average would provide a more representative summary statistic, or at least a footnote clarifying the calculation.

* Line 424-425: Provide annotation details and explain what is implied under ground-truth structured data.

* The paper would benefit from a broader discussion on how the released dataset aligns with real-world data and practical applications.

* Line 452-455: The analysis seems to slightly contradict the observed findings: the majority of the models is said to show Analysis performance decline while 2 of 3 models show the accuracy increase.

---

> ### Author Response · Authors · 2025-11-19
> **Response to W1& Q1: Add references to support the claimed weaknesses of OCSR methods**
>
> Thank you for the suggestion. We have added supporting references in our revised version, as follows:
>
> > “To systematically evaluate these capabilities, we argue that models must be assessed on visual spatial representations directly. Existing approaches that convert diagrams to text formats like SMILES through Optical Chemical Structure Recognition (OCSR) are inadequate for topological reasoning evaluation. Such text-based representations discard crucial spatial positioning, pathway directionality, and global connectivity patterns that are essential for structural analysis [1-4]. Moreover, OCSR transcription errors would further degrade reasoning performance, making it impossible to isolate and evaluate pure topological reasoning capabilities [5-7].”
>
> In addition, we now make this limitation concrete in two ways:
>
> **(1) Example of RxnScribe output and its lack of topology**
>
> We add a typical RxnScribe[3] prediction for a reaction scheme:
>
> ```json
> [
>   {
>     "reactants": [
>       {
>         "category": "[Mol]",
>         "smiles": "Cc1ncc(C[n+]2csc(CCO)c2C)c(N)n1.Cl"
>       }
>     ],
>     "conditions": [
>       {
>         "category": "[Txt]",
>         "text": "28.5% aq"
>       },
>       {
>         "category": "[Mol]",
>         "smiles": "O=S(=O)([O][Na])SCC1CCCO1"
>       },
>       {
>         "category": "[Txt]",
>         "text": "NaOH"
>       },
>       {
>         "category": "[Txt]",
>         "text": "rt, 10 min"
>       }
>     ],
>     "products": [
>       {
>         "category": "[Mol]",
>         "smiles": "C/C(=C(\\CCO)SSCC1CCCO1)N(C=O)Cc1cnc(C)nc1N"
>       }
>     ]
>   }
> ]
> ```
> This representation correctly recovers which molecules and textual labels appear in the image, but it does not encode the reaction topology that our benchmark targets:
> - it does not specify which arrow connects which reactant and product
> - it does not indicate the order of individual steps in a multistep scheme
> - it does not represent branching or convergent steps in the global reaction network
>
> As a result, a model that only sees OCSR derived SMILES or JSON cannot be evaluated on the full topological structure of the reaction scheme.
>
> **(2) Ablation: using RxnScribe JSON instead of images**
>
> To further validate this point empirically, we re-ran our benchmark by replacing the reaction diagrams with the corresponding RxnScribe JSON outputs (as above), while keeping everything else unchanged. The table below shows Qwen2.5-VL results on the four tasks (Localization, Extraction, Reasoning, Analysis):
> | Model         | Input type | Localization | Extraction | Reasoning | Analysis |
> | ------------- | ---------: | -----------: | ---------: | --------: | -------: |
> | Qwen2.5-VL 3B |      Image |        24.43 |      89.61 |     49.70 |    34.65 |
> | Qwen2.5-VL 3B |       JSON |        34.37 |      34.54 |     19.17 |     9.90 |
> | Qwen2.5-VL 7B |      Image |        37.96 |      85.75 |     66.47 |    39.11 |
> | Qwen2.5-VL 7B |       JSON |        39.88 |      41.06 |     10.78 |     8.91 |
>
> While Localization changes only mildly when switching to JSON, the scores on Extraction, Reasoning, and Analysis drop significantly. For example, for Qwen2.5-VL 3B, Extraction decreases from 89.61 to 34.54 and Analysis from 34.65 to 9.90. For Qwen2.5-VL 7B, Reasoning drops from 66.47 to 10.78. This controlled experiment confirms that OCSR derived JSON, which lacks explicit reaction topology, is not sufficient to support the higher level topological reasoning that our benchmark is designed to test. Therefore, evaluating models directly on visual reaction diagrams is necessary and consistent with our claim in Lines 87-92.
> - [1] DECIMER: Towards Deep Learning for Chemical Image Recognition, 2021.
> - [2] MolScribe: Robust Molecular Structure Recognition with Image-to-Graph Generation, 2023.
> - [3] RxnScribe: a sequence generation model for reaction diagram parsing, 2023.
> - [4] Img2Mol – Accurate SMILES Recognition from Molecular Graphical Depictions, 2021.
> - [5] Optical Structure Recognition Software To Recover Chemical Information: OSRA, An Open Source Solution, 2009.
> - [6] ChemGrapher: Optical Graph Recognition of Chemical Compounds by Deep Learning, 2020.
> - [7] DECIMER.ai: an open platform for automated optical chemical structure identificat

---

> ### Author Response · Authors · 2025-11-19
> **Response to W2 & W6 & Q3 & Q7 & Q8: More detailed description about Dataset Collection, Annotation and Release**
>
> Thank you for the suggestion. We have expanded the dataset section to describe the collection pipeline, annotation protocol, quality control, and release plan in more detail:
>
> **(1) Collection pipeline.**
>
>    We manually curated a dataset of chemical reaction images from major ACS journals, including Organic Process Research & Development (OPR\&D), The Journal of Organic Chemistry (JOC), Journal of the American Chemical Society (JACS), and Organic Letters in the period 1996-2016. The selected images span a wide range of complexity, from simple single step reactions to multistep processes with multiple branches and intermediates. For each image, we designed several diverse questions, covering both basic recognition (reactants, products, reagents, and common notations) and higher level reasoning (reaction conditions, mechanistic relations, yield related questions). In total, the dataset currently contains 1,618 question-answer pairs.
>
> **(2) Annotation guidelines and annotator qualification.**
>    Each reaction diagram is annotated with structured ground truth in JSON format, including reactants, products, reagents, and conditions. Annotators followed written guidelines that specify how to (i) identify and label reactants, products, and conditions, and (ii) write question-answer pairs that explicitly target structure, mechanism, and conditions. All annotators are graduate level chemists. Before starting large scale annotation, they were trained on pilot examples and their annotations were reviewed and corrected by a senior expert to align their understanding of the guidelines.
>
> **(3) Inter annotator agreement.**
>    Each sample was independently checked by three annotators. Disagreements were resolved through discussion, with the senior expert making the final decision when needed. Although we did not compute a formal \(\kappa\) score, the empirical consistency after review exceeded 95%. We will clarify this in the revised manuscript.
>
> **(4) Release plan.**
>    Upon acceptance, we will publicly release:
>    - all 1,618 QA pairs with ground truth answers and task annotations
>    - all 1,300+ reaction diagram images
>    - complete evaluation code, including scoring scripts, analysis tools, and error analysis utilities
>    - all prompts and experimental configurations used for model evaluation
>
>    The dataset will be released via a GitHub repository together with a permanent archive on a public platform such as Zenodo or Hugging Face Datasets within two weeks after acceptance.
>
> **(5) Practical relevance of the dataset.**
>    Our dataset is constructed from real reaction diagrams in ACS journals, so it reflects the layout variability, notation style, and information density encountered in practical chemical literature. The images cover a broad spectrum of organic reaction classes, and each diagram is associated with multiple questions that probe different aspects of understanding and reasoning. Furthermore, every image is taken from a different article to maximize diversity at the document and reaction level. We will highlight these properties in the paper to clarify how the dataset supports realistic evaluation of multimodal reasoning in chemistry.

---

> ### Author Response · Authors · 2025-11-20
> **Response to W3: Add a LLM baseline using SMILES or SELFIES as input**
>
> Thank you for raising this point. We agree that a single-modal LLM baseline with a structured textual representation of the reaction graph would help disentangle visual parsing, cross-modal alignment, and chemical reasoning.
>
> In our current dataset, however, we only have reaction schemes in image form and do not possess human curated ground truth SMILES/SELFIES or reaction graphs for all molecules and steps. Constructing such a textual graph resource would require an additional large scale OCSR plus expert curation effort, which is beyond the scope of the present work, but we view it as a valuable extension.
>
> To partially address your concern, we conducted a new experiment using the best available OCSR tool in our pipeline (RxnScribe) to convert each reaction diagram into a JSON representation. We then provided this JSON (linearized as text) to Qwen2.5-VL in text-only mode, with no images, while keeping the QA tasks unchanged. The table below compares the original image based setting with this RxnScribe JSON based setting on the four tasks (Localization, Extraction, Reasoning, Analysis):
>
> | Model           | Input type | Localization | Extraction | Reasoning | Analysis |
> |-----------------|-----------:|-------------:|-----------:|----------:|---------:|
> | Qwen2.5-VL 3B   | Image      | 24.43        | 89.61      | 49.70     | 34.65    |
> | Qwen2.5-VL 3B   | JSON       | 34.37        | 34.54      | 19.17     | 9.90     |
> | Qwen2.5-VL 7B   | Image      | 37.96        | 85.75      | 66.47     | 39.11    |
> | Qwen2.5-VL 7B   | JSON       | 39.88        | 41.06      | 10.78     | 8.91     |
>
> Localization changes only mildly when switching from images to JSON, but Extraction, Reasoning, and Analysis scores drop sharply. For example, for Qwen2.5-VL 7B, Reasoning drops from 66.47 (image) to 10.78 (JSON), and Analysis from 39.11 to 8.91. This indicates that providing a flat list of SMILES and conditions from OCSR does not make the tasks easier; instead, the model heavily relies on the visual diagram, which encodes spatial layout and reaction topology that are not present in the RxnScribe JSON.
>
> While this experiment does not replace a true ground truth textual graph baseline with a dedicated unimodal LLM, it provides a concrete reference level and supports our claim that topological reasoning over reaction diagrams cannot be reduced to reasoning over a set of OCSR derived SMILES alone. We will add this ablation and its discussion in the revised version and explicitly mention that building a human curated textual graph benchmark for unimodal LLMs is an important direction for future work.

---

> ### Author Response · Authors · 2025-11-20
> **Response to W4: Add clearer illustrations for figures**
>
> Thanks for you suggestion. We will include clearer explanations about chemistry examples in figures, especially for Fig2 and Fig4, to make the paper more accessible to readers without a chemistry background.

---

> ### Author Response · Authors · 2025-11-20
> **Response to Q2: Specify the "existing benchmarks" in Line 094**
>
> We thank the reviewer for this constructive suggestion. In the revised manuscript, we have explicitly specified that "existing benchmarks" primarily refer to datasets focused on general document understanding (e.g., **DocVQA** [1], **TextVQA** [2]) and chart analysis (e.g., **ChartQA** [3], **ScienceQA** [4]).
>
> Our statement highlights a critical distinction: while these benchmarks effectively evaluate a model's ability to perform OCR and extract semantic information (e.g., reading text or retrieving specific values), they do not strictly stress-test the **structural reasoning** capabilities required to parse complex topologies, such as the connectivity and reaction flow in chemical reaction diagrams.
>
> We have updated Line 094 to include these specific citations for clarity.
>
> **References:**
>
> [1] Mathew, Minesh, et al. "DocVQA: A Dataset for VQA on Document Images." *Proceedings of the IEEE/CVF Winter Conference on Applications of Computer Vision (WACV)*. 2021.
>
> [2] Singh, Amanpreet, et al. "Towards VQA Models That Can Read." *Proceedings of the IEEE/CVF Conference on Computer Vision and Pattern Recognition (CVPR)*. 2019.
>
> [3] Masry, Ahmed, et al. "ChartQA: A Benchmark for Question Answering about Charts with Visual and Logical Reasoning." *Findings of the Association for Computational Linguistics: ACL 2022*.
>
> [4] Lu, Pan, et al. "Learn to Explain: Multimodal Reasoning via Thought Chains for Science Question Answering." *Advances in Neural Information Processing Systems (NeurIPS)*. 2022.

---

> ### Author Response · Authors · 2025-11-20
> **Response to Q4: Clarify "incomplete mechanisms" and "ambiguous visual representations."in Line 183-187**
>
> **(1) Incomplete mechanisms.**
> For incomplete mechanisms, we refer to reaction diagrams missing essential components such as reagents, catalysts, or products, or those that show only partial transformations without clear start or end species.
>
> **(2) Ambiguous visual representations.**
> For ambiguous visual representations, we mean figures with unclear atom mappings, overlapping arrows, or inconsistent symbols that make the reaction pathway visually indeterminate.
>
> These cases were excluded to ensure all retained diagrams have well defined and interpretable chemical structures.

---

> ### Author Response · Authors · 2025-11-20
> **Response to Q6: Use the weighted average over four tasks to instand of an arithmetic average in Table 2**
>
> Following your suggestion, we compute both arithmetic and weighted averages across tasks for each model. We group models into API based and open source. For each group, the best result is shown in **bold**, and the second best in *italics*.
>
> | API based Models  | Params | Arithmetic Avg. | Weighted Avg. |
> |-------------------|:------:|:---------------:|:-------------:|
> | GPT-4o            |   –    | 63.83           | 57.79         |
> | Claude-3.5-Sonnet |   –    | **68.60**       | **62.86**     |
> | Gemini-1.5-Pro    |   –    | 62.17           | 57.29         |
> | Qwen-VL-MAX       |   –    | *65.03*         | *61.62*       |
>
> | Open source Models      | Params | Arithmetic Avg. | Weighted Avg. |
> |-------------------------|:------:|:---------------:|:-------------:|
> | InternVL2.5-MPO         |  8B    | 50.38           | 47.53         |
> | InternVL2.5-8B          |  8B    | 41.22           | 38.88         |
> | InternVL2.5-26B         | 26B    | 46.44           | 43.70         |
> | InternVL2.5-78B         | 78B    | 53.74           | 51.85         |
> | Qwen2.5-VL-3B           |  3B    | 49.60           | 44.99         |
> | Qwen2.5-VL-7B           |  7B    | *57.32*         | *53.28*       |
> | Qwen2.5-VL-72B          | 72B    | **68.66**       | **62.42**     |
> | DeepSeek-VL2-3B         |  3B    | 36.49           | 42.27         |
> | DeepSeek-VL2-16B        | 16B    | 36.05           | 39.31         |
> | LLaVA-NeXT-Mistral      |  7B    | 24.40           | 21.78         |
> | LLaVA-NeXT-Vicuna       | 13B    | 20.73           | 20.83         |
> | Phi-3.5-vision-instruct | 4.2B   | 39.75           | 36.84         |
> | MiniCPM-o-2.6           |  8B    | 50.33           | 47.59         |
>
> Here, the weighted average: is calculated by  $\text{Avg} _ {\text{weighted}} = \frac{\sum _ {i=1}^{N} n _ i \times s _ i}{\sum _ {i=1}^{N} n _ i},$ where $n _ i$ is the number of questions and $s_i$ is the score for task $i$. Meanwhile, the arithmetic average is calculated by $\text{Avg} _ {\text{arithmetic}} = \frac{1}{N}\sum _ {i=1}^{N}s _ i$.
>
> Comparing the two metrics, we find that they lead to very similar conclusions. Within each group, the ranking of models is unchanged: for API based models Claude-3.5-Sonnet remains the best and Qwen-VL-MAX the second best; for open source models Qwen2.5-VL-72B is consistently the best and Qwen2.5-VL-7B the second best. Model families also preserve their internal order (for example, InternVL2.5-78B > InternVL2.5-MPO > InternVL2.5-26B > InternVL2.5-8B under both metrics). The numerical differences between arithmetic and weighted averages are modest (typically within about 2–6 points), and much smaller than the gaps between model families. This indicates that using a simple arithmetic average in the original table does not introduce a strong bias, although we now report the weighted averages for completeness.

---

> ### Author Response · Authors · 2025-11-20
> **Response to Q9: Clarify minor inconsistent statement in Line 452-455**
>
> Thank you for pointing this out. We will revise the sentence for clarity as follows: "While the 3B model shows a slight decline in Analysis performance (−3.46%), the 7B and 72B models exhibit moderate improvements (+8.41% and +1.48%), suggesting that larger models may better leverage structural cues even under visual masking."

---

### Author Response · Authors · 2025-11-21
**Response to Reviewer eygh W5,Q5 & Reviewer TMog W1,Q4:  Evaluation under exact match and LLM-as-a-judge**

Thank you for these thoughtful comments on our evaluation protocol. We agree that strict exact-match scoring can in principle underestimate a model’s reasoning ability, and we have clarified both its limitations and our mitigation steps.

**(1) Strict exact matching and our current setup, including topology questions.**
First, for the “Structural Topology Analysis” questions, the answers are multiple choice rather than free-form conceptual descriptions, so strict matching does not penalize paraphrases. A typical example is:

> The diagram illustrates a reaction mechanism. Which term classifies the structure of the reaction pathway?
> A) Single line
> B) Multiple line
> C) Tree
> D) Graph
>
> **Format Rules:**
> - Reply only with the letter (A/B/C/D).

In this setting, the model only needs to output one of {A, B, C, D}. There is no risk that a correct conceptual answer like “linear chain with a branch” is marked wrong because of wording, since the evaluation is on the letter choice, not the phrase itself.

For many other questions, especially numeric ones, we also explicitly restrict the output format to minimize surface-form variation. For example, for numeric questions we use rules such as:

> Rules: (1) Reply only with the number (e.g., 80); (2) No symbols, units, or words.

Under this setup, we manually inspected the outputs and did not observe typical cases like `2` vs `two` or clearly correct entities that are only slightly paraphrased but marked wrong. Therefore, in our current experiments the practical impact of exact matching is limited, although we fully acknowledge that exact-match scoring is conservative and may underestimate partial understanding in principle. We thank the reviewers for pointing this out.

**(2) Additional experiment: LLM-as-a-judge with free-form answers.**
To directly probe semantic equivalence, we added a complementary evaluation protocol where we:

- remove the strict answer-format rules and allow free-form answers
- use an external LLM-as-a-judge (Gemini-2.5-Flash) to decide correctness based on semantics

For example:

- **Question:** “What is the percentage yield of the step from 5 to 4 in the diagram?”
- **Expected answer:** `98`
- **Model answer:** “The percentage yield of the step from 5 to 4 is 98%.”
- **Judge verdict:** correct.

In this setting, semantically correct but non-exact surface forms are counted as correct.

**(3) Effect on quantitative results under both evaluation settings.**
We report results for Qwen2.5-VL-3B and Qwen2.5-VL-7B on the four tasks (Localization, Extraction, Reasoning, Analysis), together with both the simple arithmetic average and a question-count weighted average.

We compare:
- exact-match with constrained answers (original evaluation)
- LLM-as-a-judge with free-form answers (new evaluation)

| Model          | Eval protocol                        | Localization | Extraction | Reasoning | Analysis | Arithmetic avg | Weighted avg |
|----------------|--------------------------------------|-------------:|-----------:|----------:|---------:|---------------:|-------------:|
| Qwen2.5-VL 3B  | Exact match, constrained answers     | 24.43        | 89.61      | 49.70     | 34.65    | 49.60          | 44.99        |
| Qwen2.5-VL 3B  | LLM-as-judge, free-form answers      | 36.40        | 83.33      | 70.06     | 25.25    | 53.76          | 50.49        |
| Qwen2.5-VL 7B  | Exact match, constrained answers     | 37.96        | 85.75      | 66.47     | 39.11    | 57.32          | 53.28        |
| Qwen2.5-VL 7B  | LLM-as-judge, free-form answers      | 54.01        | 87.44      | 87.42     | 42.57    | 67.86          | 64.58        |

Here, the arithmetic average is the simple mean over the four tasks. The weighted average is computed by weighting each task score by its number of questions and dividing by the total number of questions.

We observe that:

- reasoning scores increase substantially under the LLM-as-a-judge metric (for example, from 49.70 to 70.06 for 3B and from 66.47 to 87.42 for 7B), confirming that a semantic metric recovers some answers that were previously counted as incorrect
- both arithmetic and weighted averages increase for both models (for example, the weighted average for Qwen2.5-VL-7B increases from 53.28 to 64.58)
- the relative ranking between 3B and 7B remains unchanged, and the performance gaps are still large

This suggests that our original exact-match evaluation is conservative but does not distort the relative comparison between models. The LLM-as-a-judge results provide a more permissive semantic view that is consistent with our main conclusions.

---

> ### Author Response · Authors · 2025-11-21
>
> **(4) Why we keep exact match as the primary metric.**
> At the same time, we are cautious about using LLM-as-a-judge as the primary evaluation for the full benchmark. Recent work on LLM-as-a-judge (for example, "Preference Leakage: A Contamination Problem in LLM-as-a-judge") has pointed out potential issues such as contamination and preference leakage, where the judge may encode training-time exposure to similar content. In addition, running a judge model for every prediction introduces nontrivial computational and monetary cost for large-scale evaluation. More importantly, our current testing methodology is fully reproducible, whereas LLM-as-a-judge cannot guarantee consistent results across runs.
>
> For these reasons, in this work we keep strict exact-match scoring as the main metric, clearly acknowledge its conservative nature, and present the LLM-as-a-judge evaluation as a complementary analysis. We will update the paper to (i) explicitly discuss the limitations of exact matching, including for topology-related tasks, (ii) report the additional LLM-as-a-judge results, and (iii) highlight richer semantic evaluation and normalization as important directions for future work.

---

### Meta-Review · Area_Chair_iStK · 2026-01-06

**Summary:**

All reviews rated the paper as borderline. A consensus of concern has emerged among the reviewers regarding both the reproducibility of the reported results and the generalizability of the proposed dataset collection methodology. Additionally, the assessment of topological reasoning constitutes a shared point of contention across the reviews. Although the authors have promised to add a topology-only subset for validation, this requires further discussion and revision, which is crucial for evaluating the significance of this paper. Thus, I recommend that the paper should be revised and resubmitted with more comprehensive evaluation.

**Reviewer Concerns:**

The majority of the concerns raised have been successfully addressed. However, the evaluation regarding topological reasoning remains an unresolved core issue.

**Reviewer Scores:**

Reviewer eygh and Reviewer TMog may maintain their scores due to the aforementioned unresolved issues. Reviewer QNNg may increase the score.

---

### Decision · Program_Chairs · 2026-01-26

Reject